# Exploring the Effects of Rearing Densities on Epigenetic Modifications in the Zebrafish Gonads

**DOI:** 10.3390/ijms242116002

**Published:** 2023-11-06

**Authors:** Alejandro Valdivieso, Marta Caballero-Huertas, Javier Moraleda-Prados, Francesc Piferrer, Laia Ribas

**Affiliations:** 1IHPE, Université de Montpellier, CNRS, IFREMER, Université de Perpignan Via Domitia, 34090 Montpellier, France; 2CIRAD, UMR ISEM, 34398 Montpellier, France; marta.caballero_huertas@cirad.fr; 3ISEM, Université de Montpellier, CIRAD, CNRS, IRD, EPHE, 34090 Montpellier, France; 4Institut de Ciències del Mar, Consejo Superior de Investigaciones Científicas (ICM-CSIC), 08003 Barcelona, Spain; jmp.cyc@gmail.com (J.M.-P.); piferrer@icm.csic.es (F.P.)

**Keywords:** stress, gonad, sex, methylation, masculinization, rearing

## Abstract

Rearing density directly impacts fish welfare, which, in turn, affects productivity in aquaculture. Previous studies have indicated that high-density rearing during sexual development in fish can induce stress, resulting in a tendency towards male-biased sex ratios in the populations. In recent years, research has defined the relevance of the interactions between the environment and epigenetics playing a key role in the final phenotype. However, the underlying epigenetic mechanisms of individuals exposed to confinement remain elucidated. By using zebrafish (*Danio rerio*), the DNA methylation promotor region and the gene expression patterns of six genes, namely *dnmt1*, *cyp19a1a*, *dmrt1*, *cyp11c1*, *hsd17b1*, and *hsd11b2*, involved in the DNA maintenance methylation, reproduction, and stress were assessed. Zebrafish larvae were subjected to two high-density conditions (9 and 66 fish/L) during two periods of overlapping sex differentiation of this species (7 to 18 and 18 to 45 days post-fertilization, dpf). Results showed a significant masculinization in the populations of fish subjected to high densities from 18 to 45 dpf. In adulthood, the *dnmt1* gene was differentially hypomethylated in ovaries and its expression was significantly downregulated in the testes of fish exposed to high-density. Further, the *cyp19a1a* gene showed downregulation of gene expression in the ovaries of fish subjected to elevated density, as previously observed in other studies. We proposed *dnmt1* as a potential testicular epimarker and the expression of ovarian *cyp19a1a* as a potential biomarker for predicting stress originated from high densities during the early stages of development. These findings highlight the importance of rearing densities by long-lasting effects in adulthood conveying cautions for stocking protocols in fish hatcheries.

## 1. Introduction

Rearing density holds significant importance in aquaculture, as it directly impacts fish welfare, thereby influencing overall profitability. Inappropriate stocking densities can induce stress, leading to physiological disruptions that pose a threat to fish physiology [1,2]. Persistent exposure to chronic stressors, such as high population density, can increase susceptibility to diseases, deplete energy resources, decrease growth performance, as well as muscle and bone quality, and antioxidative capacity, and, ultimately, reduce overall performance [3,4]. The ‘General Adaptation Syndrome’ is the widely accepted concept of stress and consists of three phases [5,6]. Firstly, there is a physiological state of ‘alarm’, characterized by the production of adrenaline and cortisol. Secondly, if the stressor persists, the organism attempts to adapt and defend itself, entering a state of resistance. Finally, if the stress continues beyond the organism’s coping abilities, it enters an exhaustion phase, which may lead to irreversible dysfunction or even death [7].

Stress is defined as a coordinated series of behavioral and physiological responses to any appreciable challenge to homeostasis or allostasis [8]. As in other vertebrates, fish respond to environmental challenges with a series of adaptive neuroendocrine adjustments that are collectively termed ‘the stress response’, which in the short-term could be beneficial but a prolonged activation of the stress response may lead to immunosuppression, reduced growth, and reproductive dysfunction [9]. These deregulations in the reproductive axis are key in the development of the gonads in species in which the environment influences sexual determination [10].

The stress response is connected to the reproductive system by the brain–pituitary–gonadal axis in which neuro-endocrine interactions control multi-directionally the fish physiology [11,12,13]. In many fish species, the final sex of an individual is determined by genetic and environmental factors [14]; therefore, stress needs to be considered when studying sexual development. During sex differentiation, environmental factors have the potential to influence and modify sexual phenotypes. Therefore, genes and diverse factors in their habitat are responsible for defining the fate of differentiating gonads towards an ovary or a testis. The most well-documented abiotic factor in many fish species is temperature, responsible for masculinizing fish populations [15,16]. In contrast, a limited number of studies have investigated the impact of rearing density on the final sexual phenotype [17]. For instance, this is the case with the paradise fish (*Macropodus opercularus*) [18], some coral reef fish species (e.g., *Centropyge potteri* and *Labroides dimidiatus*) [19,20], the European and American eels (*Anguilla anguilla* and *Anguilla rostrata*) [21,22,23], the European sea bass (*Dicentrarchus labrax*) [24,25], and the zebrafish (*Danio rerio*) [26,27].

Zebrafish has become a model organism in many research areas from biomedical, developmental biology, and toxicology to aquaculture-related research [28,29,30]. Although zebrafish is a freshwater fish, its advantages make it an appropriate species for studying reproduction-related problems observed in marine fish farming [30]. Nevertheless, zebrafish presents some disadvantages as an animal model, for example, the inability to collect blood from it due to its small size or the fact that its genome is almost double the size (1412 Mb) that of other fish models like medaka (*Oryzias latipes*, 800 Mb) [29]. Zebrafish is a gonochoristic species of an undifferentiated type [31,32]. The initial indication of gonadal differentiation in zebrafish occurs approximately 10 days post-fertilization (dpf), triggering the onset of a critical period during which the gonadal fate can be influenced by environmental cues. While the duration of gonadal differentiation may vary among individuals, it is typically fully accomplished by around 50 dpf [33,34,35].

Wild zebrafish populations possess a chromosomal sex-determination system, while some ‘laboratory’ populations, due to artificial selection, are characterized by a polygenic sex-determining system, both of them environmentally influenced [36,37,38]). Environmental factors able to modulate sexual phenotype and, consequently, skew sex ratios in zebrafish are temperature [27,39,40], hypoxia [41], nutritional resources [42], infection [43,44], and density levels [45,46]. To enhance the accuracy of rearing protocols and prevent the phenomenon of masculinization caused by rearing density effects, a more precise investigation into the critical window of sex differentiation in zebrafish is imperative. Previous studies conducted by Ribas et al. [45,46] predominantly encompassed the entirety of larval development, spanning from 6 to 90 dpf. However, to establish a more refined rearing protocol and gain deeper insights, it is recommended that the focus is placed on the specific time frame of sex differentiation, which occurs between 15 and 45 dpf in zebrafish.

For more than a decade, aquaculture-related researchers have manifested the role of epigenetics in environmental conditions to better understand fish physiology to improve aquaculture production. Thus, searching for informative epimarkers is at the frontier of aquaculture-related research. The effects of temperature in altering DNA methylation levels in some canonical sex-related genes associated with the masculinization of the population have been shown in many farming fish species: in the European sea bass [47,48,49], Nile tilapia (*Oreochromis niloticus*) [50,51], turbot (*Scophthalmus maximus*) [52], half-smooth tongue sole (*Cynoglossus semilaevis*) [53], and the Japanese flounder (*Paralichthys olivaceus*) [54].

In zebrafish, exposure to high temperatures during sex differentiation has been shown to engender changes in DNA methylation on genes involved in reproduction and stress, with these changes in levels associated with the masculinization of the populations [38]. Based on these substantial differential changes, specific epigenetic markers (i.e., CpGs) have been used to predict the sex and the past-thermal events that occurred during the early stages of gonad development [55]. Moreover, in a multigenerational experiment, hypomethylation of the testicular epigenome in the unexposed first-generation offspring derived from the heat-exposed parents was observed [56]. These results confirmed that environmental disturbances may involve an epigenetic memory in which molecular biomarkers can be valuable toolkits for better understanding the environmental cues that alter phenotypes during early development.

The present study aimed to identify putative epigenetic biomarkers able to predict past stress events caused by rearing density. To this end, we used a locus-specific approach to evaluate the DNA methylation of genes involved in (1) epigenetics: DNA methyltransferase 1 (*dnmt1*), the enzyme responsible for re-establishing the methylation landscape in the cell [57]; (2) reproduction: cytochrome P450, family 19, subfamily A, polypeptide 1a (gonadal aromatase, *cyp19a1a*), the enzyme that converts androgens to estrogens [58], and the doublesex and mab-3-related transcription factor 1 (*dmrt1*), a key regulator of sex determination of testis development [59], and (3) stress: cytochrome P450 family 11 subfamily b member 1 (*cyp11c1*), the enzyme involved in the conversion of testosterone to cortisol [60], hydroxysteroid (17-beta) dehydrogenase 1 (*hsd17b1*), the enzyme involved in estrogen production [61], and hydroxysteroid dehydrogenase type 2 (*hsd11b2*), which converts cortisol into its inactive form [62]. To complete the study, the expressions of these six genes were analyzed to determine their relationship with DNA methylation. In addition, to shed light on the density effects on the rearing protocols in the zebrafish facilities, the identification of the sensible window that skews sex ratios towards males during sex differentiation was assessed.

## 2. Results

### 2.1. Effects of Elevated Density on Sex Ratio and Growth

Results showed that rearing density had a substantial effect which was dependent on the exposure period to which larvae were subjected (Appendix A). Comparing the effect of density levels (9 and 66 fish/L) in the 7–18 dpf group, we did not find differences in the proportion of males (Figure 1A). However, the impact of the 66 fish/L density level was significant when the treatments were performed during 18–45 dpf (Figure 1B, χ^2^ = 13.9378, *p* = 0.000189). In this second period, the overall percentage of males from the six families increased from (mean ± SD) 55.22 ± 9.10% at 9 fish/L to 67.54 ± 14.62% at 66 fish/L. However, when analyzing the effects of masculinization engendered by density at the individual family level, only Family #6 showed a significant increase in males (Figure 1C). In fact, with respect to this family, both periods were studied alongside the resulting masculinization in the 7–18 dpf (χ^2^ = 6.7267, *p* = 0.009498) and the 18–45 groups (χ^2^ = 3.9563, *p* = 0.046697), confirming previous results of family-specific responses. Because we did not have equal representation of all the families tested in the control and in the 18–45 dpf group, and, in the latter, only one of the six families demonstrated clearly the effects of masculinization caused by density, we investigated the masculinization effects of elevated density through a GLMM. Results showed that density had a significant effect on the masculinization rate (*p* < 0.001) (Table 1). Based on the prediction fit of the GLMM analysis, we plotted the expected masculinization in the populations of zebrafish according to the density level (Figure 1D).

Growth was inversely related to the rearing density, with sex-related differences (Figure 2 and Appendix A). Weight (mean ± SEM) in females remained unaffected (0.40 ± 0.02 g and 0.39 ± 0.01 g for 9 and 66 fish/L, respectively) in both periods of treatment, while males exposed to 66 fish/L showed a significant (*H* = 44.38, d.f. = 1, *N* = 412, *p* < 0.001) reduction (0.23 ± 0.01 g) when compared to males from the control treatment group (0.31 ± 0.01 g) during 18 to 45 dpf treatment (Figure 2A,B). Elevated density during the 18 to 45 dpf significantly decreased length in both sexes (*H* = 4.60, d.f. = 1, *N* = 224 and *H* = 44.37, d.f. = 1, *N* = 412 with *p* < 0.05 and < 0.01 for females and males, respectively); however, such an effect was not recorded during the 7–18 dpf (Figure 2C,D).

### 2.2. Methylation Patterns in Mature Gonads

For further methylation analysis, the second period was chosen due to the observed masculinization and growth results, which were not evident during the first period. Thus, ten males and females of the family #4 from the 18–45 dpf group were selected. The total number of raw paired reads obtained from sequencing was 4245.896 with a mean of 106.15 ± 48.33 per sample (Appendix A). After trimming, the number of reads was 501.24 and 171.52 for females and males, respectively, from the 9 fish/L group and 262.97 and 109.02 for females and males, respectively, from the 66 fish/L density level (Appendix A). Mapping efficiency was 68.07 ± 8.67% and efficiency conversion was 99.55 ± 0.08% from all samples (Appendix A).

We examined the DNA methylation in the promoter region of genes coding for four different steroidogenic enzymes (*cyp19a1a*, *cyp11c1*, *hsd17b1*, and *hsd11b2*), one transcription factor in sex-related development (*dmrt1*), and one DNA methyltransferase (*dnmt1*) and how the effects of elevated density during the period 18–45 dpf affected their methylation profile. Results showed that the high-density treatment (66 fish/L) caused an effect on the methylation of a single gene, *dnmt1*, with low methylation levels in the ovaries (~40%) compared to the ovaries of control fish (9 fish/L) (~60%). Nevertheless, the methylation level of this gene was very low in all the groups (<1%). The mean DNA methylation levels were significantly higher in testes for *cyp19a1a*, *cyp11c1*, *hsd17b1*, and *hsd11b2*, while the DNA methylation levels for *dmrt1* (*p* < 0.001) were lower when compared to ovaries at 90 dpf (Figure 3). These differences were due to sexual dimorphism, but the elevated density had no effect. Thus, sex-related differences in methylation were observed in all the six tested genes.

### 2.3. Gene Expression in Mature Gonads

*Dnmt1* was downregulated under the high-density treatment for both ovaries and testes, showing significant (*p* < 0.05) differences only in the latter (change of −2.5-fold) compared to the control (Figure 4B). The gene expressions of *dmrt1* and *hsd11b2* were not significantly altered for any of the sexes under the treatment, although a decrease in the expression of both genes appeared in the testes subjected to the 66 fish/L density. By contrast, the *cyp19a1a* gene was significantly downregulated (*p* < 0.001) in the ovaries of stressed fish (six-fold change) relative to females in the control group. In addition, no changes in the expression of *cyp11c1* were observed for any of the gonadal types at different densities.

### 2.4. Correlation of DNA Methylation vs. Gene Expression in Mature Gonads

We conducted correlation analyses between DNA methylation and gene expression levels for all of the genes selected (Figure 5). Negative trends in the correlation of methylation and gene expression were observed under both densities (9 fish/L and 66 fish/L) for *dmrt1*, *cyp19a1a*, and *dnmt1* in ovaries, despite a non-significant correlation. *Hsd11b2*, however, showed a positive correlation trend in ovaries under the high-density treatment (ρ = 0.75), although, again, without attaining statistical significance. In testes, negative trends in the correlation of methylation and gene expression were observed for *dmrt1*, *hsd11b2*, and *dnmt1*, although these correlations were also statistically non-significant. Despite the lack of statistical significance, *cyp11c1* showed a positive correlation trend in males under the high-density treatment for both control (ρ = 0.11) and treatment (ρ = 0.72), and *cyp19a1a* in males belonging to the high-density group (ρ = 0.24). Thus, overall, data indicated that no significant correlation was observed in any of the studied comparisons.

## 3. Discussion

Overall, our data indicated that exposing zebrafish larvae to high rearing densities (66 fish/L) during the 18 to 45 dpf was able to skew the sex ratio towards males. The beginning of the sexual differentiation in zebrafish into an ovary or a testis occurs from 20 to 25 dpf [63,64,65]. Nevertheless, Pradhan and Olson [66] described the first sign of gonadal differentiation at 10 dpf, when the primordial germ cells show signs of oogenesis. Here, the masculinization by effects of elevated density was not observed in earlier periods of development (7–18 dpf). However, the control groups exhibited a male predominance for the first period, exceeding 50% and possibly concealing any potential masculinization effects of density. Similarly, treatments with DNA methyltransferase inhibitor (5-aza-dC) altered sex ratios when larvae were exposed at 20–30 dpf, with no significant effects during earlier periods of gonadal development (i.e., 10–20 dpf [67]). In contrast, high temperature during the 7–21 dpf and 21–32 dpf developmental stages led to the masculinization of zebrafish populations [27]. This suggests that temperature might exert a more significant influence compared to population density as a stress factor. Altogether, our findings reveal that the phase of susceptibility to density-related effects occurs around the midpoint of gonad formation.

To investigate the impact of population density, we used standard 3 L tanks commonly employed in laboratories worldwide, which imposed limitations on the number of individuals per group. Consequently, our sample size remained relatively small, lacking the statistical power to detect potential differences that might have emerged with a larger sample per group, in particular the differences observed in the first period of treatment. In addition, inter-family variation in sex ratios was observed. Although the overall percentage of males increased under high density in all tested families, only one out of the six investigated families (family #6) showed a significant increase in males. The inter-family variation in zebrafish was described by Liew et al. [36], reporting the importance of the interaction between genetic and environmental (GxE) factors in this fish species. Later temperature and density experiments corroborated the existence of these inter-family variations in zebrafish [27,45,46,68]. Moreover, the importance of the ‘natural’ populations with genetic sex determination using different families has been recently highlighted, being responsible for sensitivity to the environment [38]. In light of these findings, we recommend the following strategies to enhance the reliability of zebrafish research: (1) replicate the experiment multiple times, (2) incorporate different biological replicates, and (3) combine the resulting outcomes from different zebrafish families to mitigate inter-family variation. Hence, even though a significant skew in sex ratios was not consistently observed across all tested families, it remains crucial to consider the density conditions that fish experience during their gonadal development. Consequently, this finding underscores the importance of considering density management in zebrafish husbandry protocols. Growth was inversely related to stocking density during the 18 to 45 dpf. This result was observed in our previous experiments [45] as well as in Hazlerigg et al. [26] in zebrafish and other cultured species [4,69,70]. In particular, the males showed lower growth than the females. The presence of sexual dimorphism in the growth response of those fish subjected to high density might be related to the faster growth of the females during development in zebrafish [42]. This effect might counteract the environmental cues during the sensitive window of sex differentiation.

In the last decade, the importance of epigenetics as a connection between the genotype and environmental influences has become more and more important for animal production, including aquaculture [71]. The most studied epigenetic event is DNA methylation. Nevertheless, efforts toward identifying the epigenetic marks that affect the expression of genes and how they persist throughout life still require years of research. After investigating different temperature regimes during early development in the European sea bass, the methylation patterns of 70 genes were identified as potential biomarkers, as diagnosis and prognosis were on a gene that fulfilled all the criteria, the keratin-associated protein 10–4 (*krtap10–4*) [72]. Recently in zebrafish, we have identified biomarkers able to predict past-thermal effects [38]. By using machine learning strategies, differential methylation levels on CpGs in the promoter region of key genes (i.e., *cyp19a1a* and the forkhead box L2a, *foxl2a* for the females and anti-Mullerian hormone, *amh* for males) were enough to predict whether an adult zebrafish had been subjected to high-temperature regimes during early gonadal development. To decipher the underlying epigenetic mechanisms triggered by high-density regimes, in the present study, we selected six genes involved in epigenetics (*dnmt1*), reproduction (*cyp19a1a* and *dmrt1*), and stress (*cyp11c1*, *hsd17b1*, and *hsd11b2*). Our results showed that the main alteration in methyl groups of the studied genes occurred in *dnmt1* in ovaries, an epimarker that was significantly hypomethylated in the ovaries almost two months after the density treatment had ended. DNA methylation changes occur due to the activity of DNA methyltransferases (dnmts) [73] and, thus, they play a central role in the DNA methylation mechanisms. In fish, many studies have shown the alteration of the expression of *dnmts*, although less data exist on the alteration of the DNA methylation patterns. Most of the studies revealed *dnmt* alterations in fish subject to toxic wastes in the water, for example, lead or atrazine treatments [74,75], and, to a much lesser extent, changes to their relation with the gonads. For example, *dnmt1* was significantly downregulated in ovaries and testes associated with bisphenol exposure in zebrafish and cyprinid rare minnow (*Gobiocypris rarus*), a phenomenon which was associated with a reduced global DNA methylation in the gonads [76,77]. Transgenerational disturbances after bisphenol were observed in some steroidogenic genes in the ovaries of rare minnow [78].

Together with the DNA methylation changes in the ovaries, our data showed that the exposure to the rearing stressor was contributing to a decrease in *dnmt1* transcript in female ovaries, but more sharply in the testicular tissue. A relationship between the masculinization effects of high-temperature treatments at early stages and dnmts alterations have been observed in Nile tilapia and zebrafish [27,79]. The masculinizing steroid 17α-methyltestosterone downregulated the expression of genes involved in chromatin histone modification and DNA methylation pathways in epigenetics, including *dnmt1*, in 18 to 19 dpf zebrafish gonads [80]. Zebrafish testes development occurs via programmed cell death from the initial undifferentiated ovary-like gonads [81]. The apoptosis events require a cascade of genes involved in several pathways, including p53, wnt signaling pathways, and the B-cell lymphoma/leukemia-2 (Bcl-2) family [82]. In mammals, the upregulation of *dnmt1* expression under oxidative stress-induced apoptosis via the hypermethylation of Bcl-2 family [83] might, therefore, alter the sex differentiation in the gonads. In this sense, our observations conclude that changes in the *dnmt1* methylation, together with its expression in the gonad during sex differentiation, were triggered by changes in the environment, i.e., high density.

Based on our data, no significant changes in the DNA methylation after rearing treatments of the other studied genes (*cyp19a1a*, *dmrt1*, *cyp11c1*, *hsd17b1*, and *hsd11b2*) were observed. This might be explained by the fact that the stressor was not severe enough to cause permanent DNA methylation changes in the studied genes, unlike that engendered by temperature in a similar experiment in which six out of ten genes showed significant changes in the DNA methylation patterns [38]. Another explanation for our data is the limitation of the methods used. Although MBS has been useful in several studies [38,49], the candidate gene approach method is limited to just a small part of the genome, near the coding region of the genes, and, thus, wide-genome strategies would have been required to fully assess modifications in the epigenome after rearing conditions.

In contrast, clear sexual dimorphism in the DNA methylation patterns was observed in our data. In the last years, evidence has accumulated highlighting the importance of the sexual dimorphic methylation patterns in fish. This is the case, for example, in Nile tilapia muscles [84] and zebrafish brains [85]. Sexual dimorphism was present in the methylation pattern in the zebrafish gonads, indicating that the immune system responses differed between fish sexes [86]. In fact, the methylation levels of certain genes were linked to a particular sex defining the Essential Epigenetic Marks (EEM) as those informative epigenetic marks that were essential for a particular sexual phenotype [87]. Two genes, *cyp19a1a* and *dmrt1*, fulfilled the EEM criteria, while in others it was not as clear, indicating that more information regarding sexual differences in the fish epigenome is required. In the present study, we added more information regarding the importance of the sexual differences in the DNA methylation of key genes related to reproduction and stress that might be a potential EEM.

High density during early development was able to downregulate the expression level of *cyp19a1a* in the ovaries. Our data are in accordance with those reported by Valdivieso et al. [88], whereby fish were subject to even higher densities (i.e., 74 fish/L). Other environmental factors were able to decrease the levels of *cyp19a1a* expression in fish, as it has been observed with high temperature [27,47,79] and hypoxia [41,89]. Therefore, based on all the available data, the lower expression of *cyp19a1a* in the ovaries of fish subjected to environmental stressors might be considered a good biomarker. The other studied genes did not show a significant regulation of their expression after the introduction of the stressor, with data showing no more changes in the DNA methylation patterns.

Correlation analyses of methylation and gene expression in the six studied genes indicated no significant differences in ovaries or testes. Most of the studied genes generally indicated negative relationships following the assumption that low DNA methylation of CpG-rich promoters is associated with the activation of the gene transcription machinery [90,91]. In contrast, in our data, we found positive correlation trends between DNA methylation and gene expression in *hsd11b2* in the ovaries, and *cyp19a1a* and *cyp11c1* in the testes. In a similar manner, positive correlations were observed in *hsd11b2* and *cyp11c1* in both testes and ovaries of zebrafish exposed to high temperatures [38], corroborating our findings. In fact, it is currently known that the DNA methylation patterns are more complex than originally thought, as other genomic elements rather than the promoters, such as gene body or introns, can contribute to transcriptional regulation [48,92,93].

## 4. Materials and Methods

### 4.1. Animal Rearing Conditions and Facility

Zebrafish (AB strain, ZDB-GENO-960809-7) were housed at the ‘Institut de Ciències del Mar’ (ICM-CSIC) in Barcelona. Fish were reared in a commercial water rack system (Aquaneering, San Diego, CA, USA). Husbandry and water conditions are described elsewhere [45]. Fish were fed ad libitum three times per day with zebrafish commercial food (Aqua Schwarz Gmbh, Maschmühlenweg Germany) and supplemented with brine shrimp (*Artemia nauplii*). Water quality parameters were monitored routinely to avoid any confounding environmental factors that could interfere with fish health and development. Appendix A provides the water parameters used during the experiment with no harmful fluctuations. Temperature, acidity, and conductivity were measured daily, while ammonia, nitrite, nitrate, and hardness were measured weekly using commercial kits (Sera, Heinsberg, Germany). To prevent any potential negative effects of masculinization before commencing the density experiments, the larvae belonging to control conditions were reared at a density of 9 fish per liter (fish/L) [45]. The experimental density of 66 fish/L was selected as a potential treatment to induce a skewed sex ratio towards males while ensuring minimal mortality that could hinder the representativeness of the population [45].

### 4.2. Experimental Design

To generate reliable biological replication, larvae from eight unrelated families (#1–8) originating from independent pairs were used to assess the effects of density during sex differentiation. A total of three and six families were used for the first (7–18 dpf) or the second (18–45 dpf) studied periods, respectively (Appendix A). Due to the large number of larvae required for each experiment, only one family (#6) produced enough eggs to assess the three groups altogether. For each family, to optimize the number of surviving larvae and to accomplish the desired density treatment in each tank, we calculated the proper volume to confine the larvae using a micro-perforated barrier in Aquaneering tanks (2.8 L volume, ZT280). The barrier was positioned along the *x*-axis position—which included a ruler—while the *y*- and *z*-axis remained fixed (Appendix A).

Three groups were created: 7–18 dpf, 18–45 dpf, and a control group for each of the studied periods of gonadal development (Appendix A). For the control group, tanks were set up with 9 fish/L at 7 dpf and the density level remained unchanged until 90 dpf. In the 7–18 dpf group, tanks started with larvae confined with 66 fish/L at 7 dpf, and larvae remained confined until 18 dpf. At this point, the physical barrier was removed to create a non-masculinization density effect that would not lead to masculinization (specifically, below 16 fish/L) [45]. In the 18–45 dpf group, the larvae were initially maintained at a density of 9 fish/L until 18 dpf, after which they were confined at 66 fish/L until 45 dpf. From 45 dpf onwards, the barrier was removed, and fish were kept at a density below 16 fish/L until 90 dpf. Due to the large number of fish per tank, if it was not possible to achieve the target density of 16 fish/L or lower at the end of the period of treatment, larvae were evenly distributed into tanks to avoid any effect of density. During the experiment, fish survival was recorded, and the barrier was adjusted as needed to maintain density at 66 fish/L. At 90 dpf, fish of all groups were euthanized using iced water. Body weight (BW ± 0.05 g) and standard length (SL ± 0.01 cm) were recorded. Fish gonads were recorded for sex ratio analysis and gonads were kept at −80 °C for further molecular analysis.

### 4.3. DNA and RNA Extractions

Molecular analyses were performed for the second period of treatment (18–45 dpf) and for the family #6. Genomic DNA and total RNA were isolated from the same gonad sample to allow comparisons of DNA methylation and gene expression in the same individual. For the DNA extraction, the gonads were treated overnight at 60 °C with a digestion buffer containing 1 μg of proteinase K (P2308, Sigma-Aldrich, St. Louis, MO, USA). Then, a standard phenol–chloroform–isoamyl alcohol protocol (PCI 25:24:1, *v*/*v*/*v*) with 0.5 μg ribonuclease A (12091021, PureLink RNase A, Life Technologies, Carlsbad, CA, USA) was implemented to eliminate RNA traces. Five hundred nanograms of DNA per sample were bisulfite-converted using the EZ DNA Methylation-Direct™ Kit (ZymoResearch, Irvine, CA, USA; D5023). For the RNA extraction, sampling was conducted using Trizol Reagent (T9424, Sigma-Aldrich, St. Louis, MO, USA) according to the manufacturer’s instructions and RNA. Quality and quantity of DNA and RNA were measured by NanoDrop (ND-1000) spectrophotometer and Qubit (Thermo Fisher Scientific, Waltham, MA, USA).

### 4.4. Methylation Bisulfite Sequencing Analysis

Gonadal DNA methylation levels were studied by Methylation Bisulfite Sequencing (MBS) following the procedures described elsewhere [38,44,49]. The studied genes were epigenetic-related (*dnmt1*), reproduction-related (*dmrt1* and *cyp19a1a*), and stress-related (*cyp11c1*, *hsd17b1*, and *hsd11b2*). The selected region for each gene included the promoter, the first exon, and the first intron extended as much as possible. The targeted portion of approximately 500 bp was amplified with PCR by using designed primers from Valdivieso et al. (2023) (Appendix A). Adaptor sequences for 16 S metagenomic library preparation (Illumina) were added to the 5ʹ ends of the primers designed: forward-TCGTCGGCAGCGTCAGATGTGTATAAGAGACAG and reverse-GTCTCGTGGGCTCGGAGATGTGTATAAGAGACAG. PCR products were indexed by Nextera XT index Kit Set A (Illumina; FC-131–2001) according to Illumina’s protocol for 16 S metagenomic library preparation and were pooled in an equimolar manner to obtain a single multiplexed library which was sequenced in a MiSeq (Illumina, San Diego, CA, USA) using the paired-end (PE) reads 250 bp protocol at the National Center of Genomic Analysis (CNAG, Barcelona, Spain). 

### 4.5. Bioinformatic Analysis

The adapters were removed using Trim galore! software (v. 0.4.5, Cambridge, UK) [94] and those reads with low-quality filtered (Phred score < 20) were discarded. To ensure that the adapters were cut off correctly, pre- and post-trim quality control was carried out using MultiQC and FastQC [95,96]. Bismark software (v.20.0, Cambridge, UK) was used to generate an in silico bisulfite-converted zebrafish genome (GCF_000002035) using the ‘bismark_genome_preparation’ function. Deduplication and methylation extraction at CpG_context were conducted using the ‘deduplicate_bismark’ and ‘bismark_methylation_extractor’ functions, respectively. Bisulfite conversion efficiency was calculated with a minimum threshold of 99.0% to accept the sample for methylation analysis. The coordinates of the CpG sites of the targeted genes were assessed through the ‘bedr’ package (v. 1.0.4) [97]. From the targeted CpG sites, only those that showed coverage > 5 times were retained for the methylation analysis.

### 4.6. Gene Expression by Quantitative PCR

For each sample, 200 ng of RNA were treated with DNAse I, Amplification Grade (Thermo Fisher Scientific Inc., Wilmington, DE, USA), and retrotranscribed to cDNA with SuperScript III RNase Transcriptase (Invitrogen, Carlsbad, CA, USA) with Random hexamer (Invitrogen, Carlsbad, CA, USA). Quantitative PCR (qPCR) was carried out in technical triplicates for each sample with the SYBR Green chemistry (Power SYBR Green PCR Master Mix; Applied Biosystems, Waltham, MA, USA). The conditions in the thermocycler were: 50 °C for 2 min, 95 °C for 10 min, followed by 40 cycles of 95 °C for 15 s and 60 °C for 1 min in a 384-well plate (CFX-386, Touch BioRad, Berkley, CA, USA). Finally, a temperature-determining dissociation step was performed at 95 °C for 15 s, 60 °C for 15 s, and 95 °C for 15 s at the end of the amplification phase. The dissociation step, primer efficiency curves, and PCR product sequencing confirmed the specificity for each primer pair. The six qPCR primers used together with the two reference genes are shown in Appendix A.

### 4.7. Statistical Analyses

All statistical analyses were performed using R software (v. 4.1.1, Vienna, Austria) [98]. Data were expressed as mean ± SEM and the differences were considered significant when *p* < 0.05. Graphs were generated using the “ggplot2“ package (v. 3.1.0) [99]. The sex ratio analysis was calculated using the chi-square (χ^2^) test with the application of the Yates correction [100] for each family. The overall analysis of density for the second period (18–45 dpf) was assessed using Generalized Linear Mixed Models (GLMMs). Density level (continuous variable) was considered the fixed factor and family was the random factor, with the response variable being the proportion of males and females. For biometry data, normality was checked with the Kolmogorov–Smirnov test and logarithmic transformations were applied, when necessary. The homoscedasticity of variances was checked with the Levene’s test. Means were compared by one-way analysis of variance (ANOVA) with Tukey’s post-hoc multiple-range test.

To obtain the DNA methylation across all the CpG sites in the studied gene region, the DNA methylation level for each site was averaged from all samples in each group. The effects of density on DNA methylation were evaluated by two-way ANOVA followed by a Tukey’s HSD test. The normality of the residuals was checked using the Shapiro–Wilk test, while the Levene’s test was used to check for homogeneity of variance.

Data obtained from qPCR were collected by SDS 2.3 and RQ Manager 1.2 software (Waltham, MA, USA). For each sample, the relative quantity (RQ) values of the genes of interest were used for normalization against the geometric mean value of two reference genes validated for zebrafish [101] and the fold change was calculated using the 2ΔΔCt method [102]. Two-way ANOVA was used to detect differences in gene expression between treatments (sex and density), after checking for homoscedasticity with the Levene’s test for every single group, as well as for normality using the Shapiro–Wilk test for each group. When normality was not followed, a Kruskal–Wallis test was performed. A Tukey’s test was used to perform *post hoc* multiple comparisons. DNA methylation and gene expression correlation analyses were carried out by running a Spearman’s rank correlation (ρ) test using the ‘corrplot’ package (v. 0.84) [103].

## 5. Conclusions

This study confirms the impact of continued exposure to stressors during gonadal differentiation on DNA methylation, gene expression, and final sexual phenotypes. Present data revealed that the sensible window for high-density occurred between 18 and 45 dpf, skewing sex ratios towards males. Among the six studied genes related to epigenetics, reproduction, and stress, *dnmt1* showed significant differences in the DNA methylation, which, if confirmed in independent studies, could constitute a suitable epimarker in the male fish subjected to stressors in the two previous months. Similarly, the inhibition of *cyp19a1a* gene expression could be considered a promising biomarker in the ovaries of fish subjected to rearing densities during sex differentiation. This analysis shed light on the characteristics of the fish reproductive system and the environment’s role in modulating epigenetic and gene expression molecular events. Thus, caution should be practiced with respect to rearing protocols in fish facilities, particularly those that use zebrafish as an animal model for research.

## Figures and Tables

**Figure 1 ijms-24-16002-f001:**
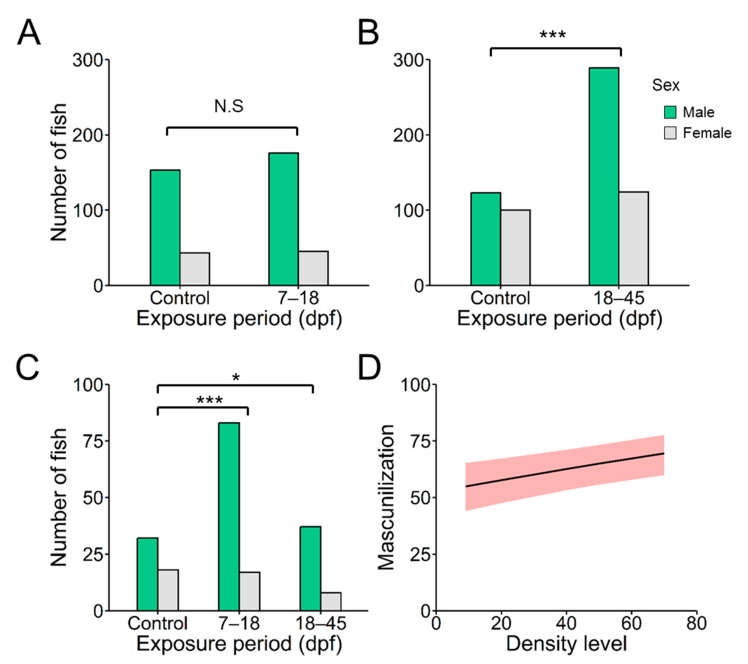
The number of males observed at different exposure windows: (**A**) 7–18 and (**B**) 18–45 days post-fertilization (dpf) at two stocking densities (9 and 66 fish/L). Sex ratio analysis between density treatments was conducted using the χ^2^ test. (**C**) The number of males observed in family #6 tested in both periods, 7–18 and 18–45 dpf. (**D**) The expected masculinization in the populations of zebrafish according to density level (number of fish/L) based on the prediction fit of a generalized linear mixed model for the 18–45 dpf period. The shadow represents the masculinization predicted by the model based on the rearing density. Abbreviations: N.S. = no significant; * = *p* < 0.05; *** = *p* < 0.001.

**Figure 2 ijms-24-16002-f002:**
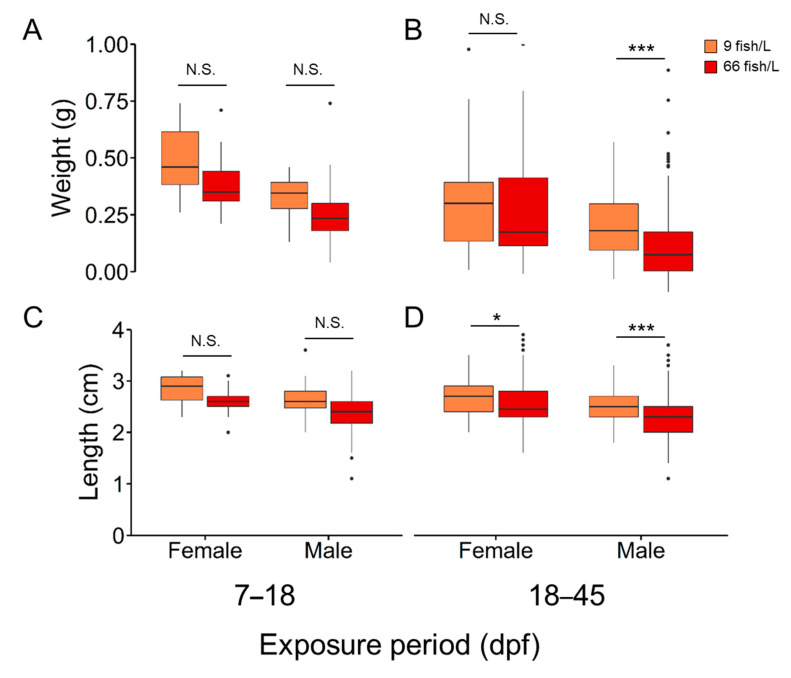
(**A**,**B**) Body mass (weight in g) and (**C**,**D**) length (cm) of females and males at different exposure windows: 7–18 and 18–45 days post-fertilization (dpf) at two stocking densities (9 fish/L and 66 fish/L). The total number of fish used was 1003; 369 (control), 221 (7–18 dpf), and 413 (18–45 dpf). Abbreviations: N.S. = no significant; * = *p* < 0.05; *** = *p* < 0.001.

**Figure 3 ijms-24-16002-f003:**
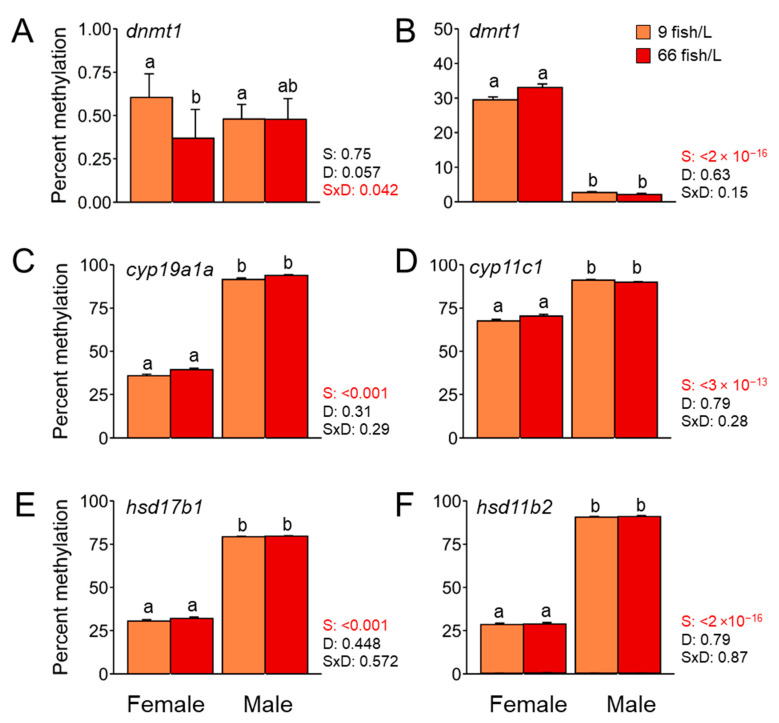
Mean DNA methylation levels in the promoter region of (**A**) *dnmt1*, (**B**) *dmrt1*, (**C**) *cyp19a1a*, (**D**) *cyp11c1*, (**E**) *hsd17b1*, and (**F**) *hsd11b2* in mature gonads of females and males (90 days post-fertilization, dpf) exposed to two rearing densities (9 and 66 fish/L) from the 18–45 dpf group. The number of fish analysed in each group was *n =* 10. Two-way ANOVA followed by a post hoc Tukey test were applied. The *p*-values for the factor effects of sex (S), density (D), and the interaction of both factors (S × D) are reported for each gene. A robust non-parametric two-way ANOVA with trimmed means was applied when data did not follow normality. Data are shown as mean ± S.E.M. Different letters indicate significant differences (*p* < 0.05) between sex and/or density.

**Figure 4 ijms-24-16002-f004:**
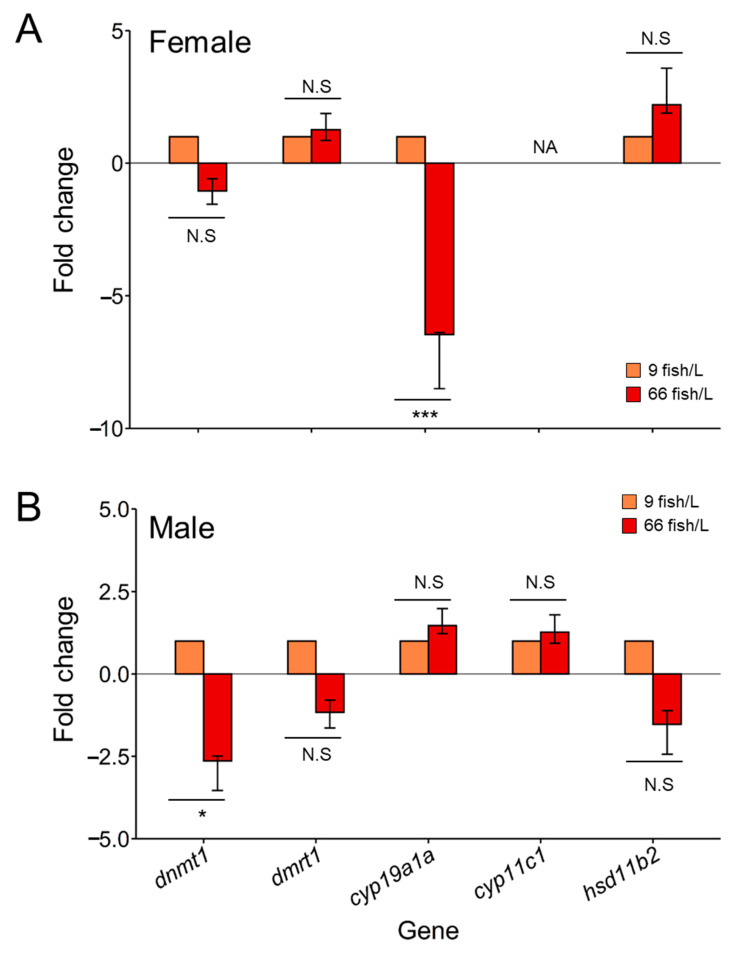
Gene expression of *dnmt1*, *dmrt1*, *cyp19a1a*, *cyp11c1*, and *hsd11b2* in (**A**) ovaries and (**B**) testes 90 days post-fertilization (dpf) after the exposure to two rearing densities (9 and 66 fish/L) from 18 to 45 dpf. Data shown as mean ± SEM. Fold change values of control group (9 fish/L) was set up at 1 as reference. Sample size: *n* = 10 (ovaries) and *n* = 10 (testes) in each group. Significant differences between sexes were analyzed using the student’s *t*-test. Abbreviations: N.S. = no significant; * = *p* < 0.05; *** = *p* < 0.001.

**Figure 5 ijms-24-16002-f005:**
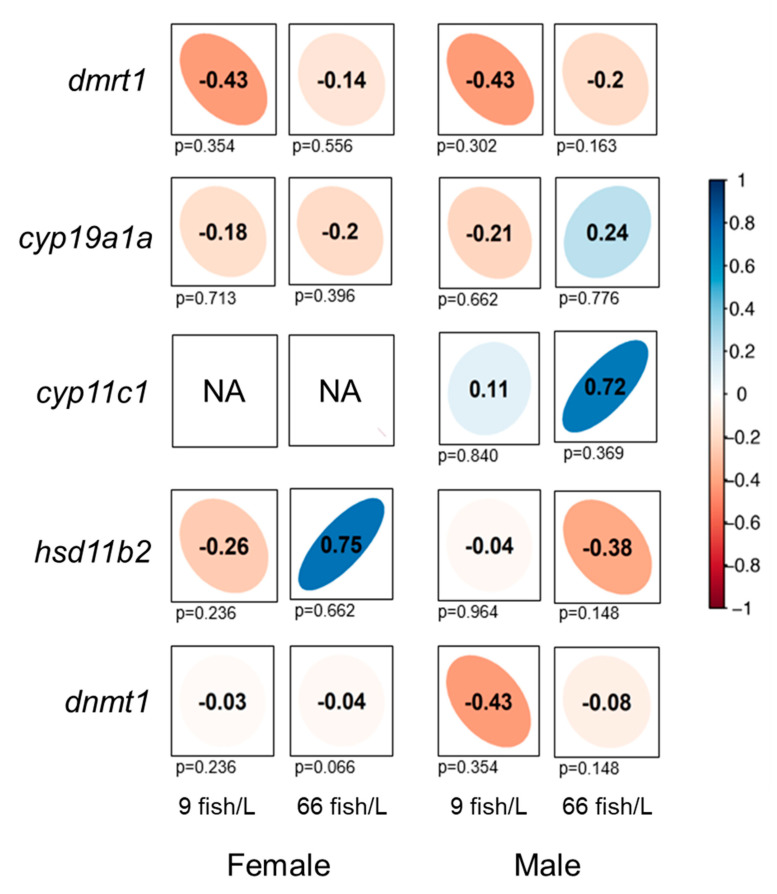
Correlations of DNA methylation of the promoter regions and gene expression levels for *dmrt1*, *cyp19a1a*, *cyp11c1*, *hsd11b2*, and *dnmt1* in the gonads of females and males (90 days post-fertilization, dpf) exposed to 9 and 66 fish/L during 18–45 dpf. Spearman’s rank correlation coefficients (ρ) are shown. The direction of the long axis of the ellipses and the color indicate the type of correlation: negative is shown in red and positive is presented in shades of blue. The short axis of the ellipse and the intensity of the color are proportional to the correlation coefficients. Significant correlations were considered when *p* < 0.05. We represented the gene *cyp11c1* in females at both densities as ‘NA’ due to the absence of gene expression data.

**Table 1 ijms-24-16002-t001:** Generalized linear mixed model (GLMM) with logit link function to test the effects of elevated density on the masculinization of zebrafish during the 18–45 days post fertilization (dpf). The density factor (9 and 66 fish/L) was set as fixed whereas family was used as the random factor effect.

Fixed Effects	Coefficient	S. E. ^a^	Z-Value ^b^	Pr(>|Z|) ^c^
Intercept	0.102	0.237	0.430	0.667
Density	0.010	0.003	3.244	0.001

Density 9 fish/L set as reference (intercept). Number of observations = 12, groups = 6. Intercepts of the random effects of the variance and standard deviation for #Family = 0.176 and 0.419, respectively. ^a^ Standard error of parameter estimate. ^b^ Z-value estimate to standard error ratio. ^c^ Pr (>|Z|) statistic for Z-value.

## Data Availability

Raw sequencing data were submitted to the Gene Expression Omnibus, with accession number GSE134646.

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
