# Peer review of "Exploring the Effects of Rearing Densities on Epigenetic Modifications in the Zebrafish Gonads"

_ijms, 2023, doi:10.3390/ijms242116002_

Round 1

Reviewer 1 Report

Comments and Suggestions for Authors

The manuscript titled “Exploring the effects of rearing densities on epigenetic modification in the zebrafish gonads” examines the effect of rearing density on the zebrafish masculinization at the adult stage. Moreover, the manuscript examines different genes due to DNA methylation, to propose possible biomarkers that will predict stress and will be applied to aquaculture hatcheries. The manuscript is well written and the experiments are well designed. The results are very innovative for zebrafish studies, but as far as I can understand, there are the first step to examine such markers before moving to applied methodologies. The title and the abstract are well aligned with the main purpose of the text. Based on these, this manuscript can be accepted for publication, after revisions that are described below (based on the given order due to manuscript format):

Major

·       In Results section, the paragraph numbering begins with 2.3. Where are the first 2 paragraphs (2.1, 2.2.)?

·       In-text there are references to supplementary figures. However, no such figures were given. The document with the supplementary figures should be provided.

·       In Material and Methods section (§4.2), the manuscript mentions about routinely monitoring abiotic factors. No data about that are given. Since these factors can affect significantly the final phenotype (especially when taking place during the early developmental stages), a table should be provided, with the main abiotic factors throughout the whole experimental period (e.g., temperature, oxygen saturation, nitrate, nitrite etc.) in order to be sure that no fluctuations were observed.

Minor

·       In §2.3, there is no mention in Fig. 1D, however its data are given. Also no caption about 1D is given in the Figure legend.

·       In table S4 the average value (with the SEM) should be added (like S5 table), based on the results that are given in text.

·       Like Fig. 1D legend, the same caution should be taken for Fig. 2D. Again, the legend for the 2D is missing.

·       In Figure 3 Legend: replace “groupis” with “group is”.

·       In the discussion section replaced the text “[…] an epimarker that were significantly hypomethylated […]” with “[…] an epimarker that was significantly hypomethylated […]”.

·       In the discussion section replaced the text “[…] fish subjected to toxic insults in the water” with the “fish subjected to toxic wastes in the water”.

·       In §4.4 there is a mention in Table S1. However, the table is wrong, since the text is referred to Table S2.

·       In §4.6 the Table S2 mention is wrong. Based on the results the correct reference is Table S3.

·       The supplementary materials text has wrong legends.

·       Some articles that are given in the reference list are not referred in text (e.g., Ambrosi C, 2017; Best C, 2018; Dimitriadi A, 2018; Edwards JR, 2017; El-Sayed, 2002). The manuscript should be revised based on that comment. All the references in the reference list, should be checked thorough and every other that is not referred in-text, should be removed.

·       In Table S1 the percentage of the Control #1 Family (35.9%) is not the correct one. 15 males out of 41 individuals is 36.58%. Examine the rest of the percentages, in case similar miscalculations are observed.

Author Response

Reviewer 1

The manuscript titled “Exploring the effects of rearing densities on epigenetic modification in the zebrafish gonads” examines the effect of rearing density on the zebrafish masculinization at the adult stage. Moreover, the manuscript examines different genes due to DNA methylation, to propose possible biomarkers that will predict stress and will be applied to aquaculture hatcheries. The manuscript is well written and the experiments are well designed. The results are very innovative for zebrafish studies, but as far as I can understand, there are the first step to examine such markers before moving to applied methodologies. The title and the abstract are well aligned with the main purpose of the text. Based on these, this manuscript can be accepted for publication, after revisions that are described below (based on the given order due to manuscript format):

Thank you so much for your time reviewing the manuscript (ms). We are glad that you find our ms interesting. Your inputs are very valuable to improve our ms before publication.  

Major

  • In Results section, the paragraph numbering begins with 2.3. Where are the first 2 paragraphs (2.1, 2.2.)?

Thanks, the numbering has been amended.

  • In-text there are references to supplementary figures. However, no such figures were given. The document with the supplementary figures should be provided.

We are sorry to hear that the supplementary figures were not accessible. This article contains supplementary figures. We have uploaded again in to the system.

  • In Material and Methods section (§4.2), the manuscript mentions about routinely monitoring abiotic factors. No data about that are given. Since these factors can affect significantly the final phenotype (especially when taking place during the early developmental stages), a table should be provided, with the main abiotic factors throughout the whole experimental period (e.g., temperature, oxygen saturation, nitrate, nitrite etc.) in order to be sure that no fluctuations were observed.

We have provided a supplementary table S4 with the abiotic parameters (temperature, acidity, conductivity, ammonia, nitrite, nitrate, and hardness)  analyzed in the water during the experiment. No harmful fluctuations were observed.  

Minor

  • In §2.3, there is no mention in Fig. 1D, however its data are given. Also no caption about 1D is given in the Figure legend.

Thanks. Figure 1D is mentioned in the text and the legend of figure 1 has been amended.

  • In table S4 the average value (with the SEM) should be added (like S5 table), based on the results that are given in text.

Table S2 (previous Table S4) has been amended by adding the average value for each group

  • Like Fig. 1D legend, the same caution should be taken for Fig. 2D. Again, the legend for the 2D is missing.

Yes, thanks, figure 2D is now added in the legend.

  • In Figure 3 Legend: replace “groupis” with “group is”.

Text has been amended.

  • In the discussion section replaced the text “[…] an epimarker that weresignificantly hypomethylated […]” with “[…] an epimarker that was significantly hypomethylated […]”.

Text has been amended.

  • In the discussion section replaced the text “[…] fish subjected to toxic insultsin the water” with the “fish subjected to toxic wastes in the water”.

Text has been amended.

  • In §4.4 there is a mention in Table S1. However, the table is wrong, since the text is referred to Table S2.

Text has been amended.

  • In §4.6 the Table S2 mention is wrong. Based on the results the correct reference is Table S3.

Text has been amended accordingly.

  • The supplementary materials text has wrong legends.

Thanks, the supplementary tables were not acurate. Legends of the supplementary tables were corrected.

  • Some articles that are given in the reference list are not referred in text (e.g., Ambrosi C, 2017; Best C, 2018; Dimitriadi A, 2018; Edwards JR, 2017; El-Sayed, 2002). The manuscript should be revised based on that comment. All the references in the reference list, should be checked thorough and every other that is not referred in-text, should be removed.

References have been checked and corrected in the ms.

  • In Table S1 the percentage of the Control #1 Family (35.9%) is not the correct one. 15 males out of 41 individuals is 36.58%. Examine the rest of the percentages, in case similar miscalculations are observed.

Table S1 has been modified by adding a column with the total number of individuals and providing the percent formula. All the percentage were doublecheck being only the one detected by the referee incorrect.

Reviewer 2 Report

Comments and Suggestions for Authors

Dear Authors,

In my opinion, before publishing your manuscript you should correct two aspects:

1) "Water quality parameters were monitored routinely in order to avoid any other environmental factor that could interfere with fish health and development."

This is not enough. 

Please add water parameters (for example in a table) and add methods that were used for measurements. How often water parameters were tested?

2) You wrote about the advantages of zebrafish, but you did not mention the disadvantages (limitations) of this model. For example, the inability to collect blood in the amount necessary for hematological and biochemical analyses. Please discuss this topic, at least in one sentence.

The manuscript is generally well-written. I especially appreciate the high-quality graphs and detailed description of the statistical method that was used. The discussion and conclusions parts are very well-written, clear and coherent.

I have no other comments. 

Good luck in corrections!

Author Response

Dear Authors,

In my opinion, before publishing your manuscript you should correct two aspects:

1) "Water quality parameters were monitored routinely in order to avoid any other environmental factor that could interfere with fish health and development."

This is not enough. 

Please add water parameters (for example in a table) and add methods that were used for measurements. How often water parameters were tested?

We have provided a supplementary table S4 with the abiotic parameters (temperature, acidity, conductivity, ammonia, nitrite, nitrate, and hardness)  analyzed in the water during the experiment. No harmful fluctuations were observed.  

2) You wrote about the advantages of zebrafish, but you did not mention the disadvantages (limitations) of this model. For example, the inability to collect blood in the amount necessary for hematological and biochemical analyses. Please discuss this topic, at least in one sentence.

Yes, it is correct. Working with zebrafish is not all advantages. The text has been amended accordingly.

The manuscript is generally well-written. I especially appreciate the high-quality graphs and detailed description of the statistical method that was used. The discussion and conclusions parts are very well-written, clear and coherent.

I have no other comments. 

Good luck in corrections!

Thanks for the revisions and your time. We are glad the you found our manuscript interesting.

Round 2

Reviewer 1 Report

Comments and Suggestions for Authors

Every comment has been taken into account and the manuscript is properly revised. 

A small minor correction

The sections 2.3-2.5 should be renamed to 2.2-2.4